# Benefits of Cardio-Pulmonary Rehabilitation in Moderate to Severe Forms of COVID-19 Infection

**DOI:** 10.3390/healthcare10102044

**Published:** 2022-10-17

**Authors:** Clara Douin, Kevin Forton, Michel Lamotte, Alexis Gillet, Philippe Van de Borne

**Affiliations:** 1Department of Cardiology, MontLegia Clinic, 4000 Liège, Belgium; 2Department of Cardiology, Erasme Hospital, 1070 Brussels, Belgium

**Keywords:** COVID 19, CPET, cardio-pulmonary rehabilitation, HIIT

## Abstract

Our aim was to evaluate the benefits of cardio-pulmonary rehabilitation on severe to moderate COVID-19 patients. 25 discharged COVID-19 patients underwent a cardio-pulmonary test (CPET), a spirometry test and a measure of carbon monoxide lung diffusion capacity (DLCO) at the beginning of their rehabilitation program and after 23 ± 5 rehabilitation sessions. This rehabilitation program combined interval training exercises on a bike and resistance exercises for major muscle groups. We then compared their progress in rehabilitation to that obtained with cardiac patients. At the beginning of their rehabilitation program, COVID-19 patients presented a reduced physical capacity with a maximal aerobic capacity (VO_2_ max) at 71% of predicted value, a maximal workload at 70% of predicted value and an exercise hyperventilation measured by a higher VE/VCO_2_ slope. Exercise was mainly limited by muscle deconditioning. After rehabilitation, the VO_2_ max and maximal workload increased in COVID 19 patients by 18% and 26%, respectively. In patients with ischemic heart disease the post-rehabilitation gains in VO2 max and maximal workload were 22% and 25%, respectively. Moreover, exercise hyperventilation decreased by 10% in both groups. On the other hand, the intrinsic pulmonary function of COVID 19 patients improved following natural recovery. In conclusion, even if cardio-pulmonary rehabilitation is probably not the only parameter which explains the partial recovery of moderate to severe COVID-19 patients, it certainly helps to improve their physical capacity and reduce exercise hyperventilation.

## 1. Introduction

In the last two years, health care professionals have taken care of patients suffering from COVID-19 infection.

There have been 618,740,522 confirmed cases of COVID-19 since the beginning of the pandemic up to October 2022, and 6,536,949 deaths from COVID-19 during the same period, according to the World Health Organization [1].

The main symptoms of COVID-19 infection are fever, fatigue, dyspnea, cough, throat soreness, headache, conjunctivitis and gastrointestinal issues [2]. The severe acute respiratory syndrome coronavirus 2 (SARS-CoV-2) primarily infects the respiratory system, and the effects can range from asymptomatic to severe acute respiratory distress syndrome (ARDS). The pathophysiological mechanisms are still under investigation, and the causes of a negative progression to ARDS are not yet fully established [3]. Additionally, it is known that the presence of preexisting cardiovascular disease increases mortality in patients with COVID-19. Conversely, cardiovascular complications, including myocarditis, cardiac rhythm abnormalities, endothelial cell injury, thrombotic events, and myocardial interstitial fibrosis, are observed in patients with COVID-19, and are associated with poor prognosis [4,5].

The present identification of patients with post-acute COVID-19 syndrome (PACS), also known as long COVID, is those with persistent symptoms, or delayed complications 4 weeks beyond the onset of the infection [6]. In patients with PACS, many continue to complain about dyspnea and/or decreased exercise capacity [6,7].

In this study, cardiopulmonary exercise testing (CPET) was used to assess the exercise capacity of patients who had previously been hospitalized for COVID-19 infection and to objectify their limitation to effort. These results also permitted the development of an individualized rehabilitation program, such as the cardiac rehabilitation program used in our center.

It is worthwhile remembering that the World Health Organization and experts recommend a rehabilitation program for patients who have suffered from COVID-19 infection, but data are still lacking regarding the modality of training, the expected objective results and the safety of such rehabilitation [8].

The aim of this study was to evaluate the benefits of cardio-pulmonary rehabilitation with high intensity interval training (HIIT) on severe to moderate COVID-19 patients. These benefits were then compared with those obtained with the patients suffered from ischemic cardiopathy, for whom the value of cardiopulmonary rehabilitation is well established.

## 2. Materials and Methods

### 2.1. Study Design

This is a single center retrospective study, approved by the local ethical committee (Erasme Ethical Committee number: B4062020000188).

Twenty-five patients who had suffered from moderate to severe forms of COVID-19 infection (meaning that they had been hospitalized in our institution for oxygen therapy during a minimum stay of one week or spent time in the intensive care unit (ICU)) were paired with twenty-five ischemic cardiopathy patients for sex, age, weight, height and initial aerobic exercise capacity. These cardiac patients had undergone cardiac rehabilitation in our center and were used as a control group.

All COVID-19 patients began cardio-pulmonary rehabilitation one month after discharge. After 2 or 3 familiarization sessions, they underwent a first CPET and the cardio-pulmonary rehabilitation program was initiated. On the 25 COVID-19 patients, 17 had also a pulmonary function testing before the rehabilitation. The CPET and the pulmonary function testing were also repeated after an average of 25 rehabilitation sessions (during a follow-up period of 10 weeks).

### 2.2. Pulmonary Function Testing

The pulmonary function testing was performed in a standardized manner according to the consensus guidelines [9,10,11]. It included dynamic spirometry, whole-body plethysmography and carbon monoxide lung diffusion capacity measurements (in 10 s) (DLCO) (Platinium Elite, MGC Diagnostics, San Paul, MN, USA).

### 2.3. CPET

CPET was performed on a cycle ergometer (Ergoselect II 1200; Ergoline, Bitz, Germany) with a step-by-step increase in load mode. The rate of work increment (W·min^−1^) was identified on an individual basis according to expected exercise tolerance and resting functional data.

VO_2_ (oxygen uptake), VCO_2_ (carbon dioxide output) and ventilation (VE) were collected breath by breath through a facial mask and analyzed every 8 s using a metabolic system (Exp’Air^®^, Medisoft, Dinant, Belgium) calibrated with room air and standardized gas. CPET was considered as maximal when two of the following criteria were met: VO_2_ increase less than 100 mL/min while workload increases further, respiratory exchange ratio (RER) above 1.10, achievement of age predicted maximal heart rate, ventilator reserve less than 15%, incapacity to maintain the pedal rate above 50 revolutions per minute.

Arterial oxygen saturation (SpO_2_) was continuously measured by finger pulse oximeter (SenSmart 8100S Serie; Nonin, MN, USA). Heart rate was (HR) obtained by 12-lead electrocardiogram (ECG) (Strässle & Co. DT 100, Albstadt, Germany) and blood pressure (BP) (Medisoft Ergoline 4M, Dinant, Belgium) was measured at the end of each stage.

The breathing reserve was calculated as the ratio between the maximal ventilation and the maximal voluntary theorical ventilation (MVV) (calculated by 35x 1-s forced expiratory volume (FEV1)).

The ventilatory threshold 1 (VT1) was determined by V-slope methods and reviewed by two blinded independent exercise physiologists.

The VE/VCO_2_ slope was determined using linear regression analysis of VE and VCO_2_ obtained throughout the exercise period.

Wassermann prediction equations for VO_2_ max were used as (0.032 × height (cm)) − 0.024 × age) + 0.019 × weight (kg) − 3.17 for men and (0.032 × height (cm)) − 0.024 × age) + 0.019 × weight (kg) − 0.49 − 3.17 for women; and were used for prediction equations for workload: (20.4 × height (cm)) − 8.74 × age − 1909)/6 for men and (20.4 × height (cm)) − 8.74 × age − 208 − 1909)/6 for women.

The limiting factors of exercise capacity were considered. The cause of limitation to exercise was determined for all participants with VO_2_ peak < 80% predicted. Pulmonary limitation to exercise was considered when the breathing reserve was <15% and/or when progressive peripheral oxygen desaturation was found. Circulatory limitation was considered when the Wassermann flowchart led to a circulatory category, including reduced oxygen pulse with a flattening (or downward) curve during incremental exercise, ECG changes with apparition of ischemia or arrhythmia, chronotropic insufficiency or a drop in blood pressure during exercise. Deconditioning was considered in participants with VO_2_ peak < 80% predicted without evidence of ventilatory or circulatory exercise limitations and with early VT1 and some degree of hyperventilation.

### 2.4. Cardiopulmonary Rehabilitation

The program of the proposed cardio-pulmonary rehabilitation comprised dynamic exercises based on high intensity interval training (HIIT) on an ergometer bike and resistance exercises on the major muscle groups. The cardio-pulmonary rehabilitation program was supervised by a cardiologist in cardiac rehabilitation and physiotherapists. The sessions were organized 3 times a week in groups of a maximum of 8 patients, in a quiet room, at stable temperature, for duration of maximum 1 h 30 min. All the patients wore facial masks during the training session according to national law.

The HIIT part of the training consisted of 6 min warm-up period, followed by a block of interval exercises: 2 min at high intensity (workload at 80% of the Watt max during CPET) and 4 min at low intensity (workload at 50% of the Watt max) with a duration of 24 min. This was followed by a cool-down period of 6 min.

For the resistance exercises, the one repetition maximal (1 RM) was determined after the familiarization sessions on the vertical traction and triceps machine for upper limb and the leg press, the leg extension, the leg curl for lower limb (main machines used). To determine the one repetition maximum, we asked the patient to perform the maximum number of repetitions with a weight that was gradually increased by 2 kg for the upper limbs and 5 kg for the lower limbs. The one repetition maximal was the load with which the patient was able to do only one repetition. The resistance exercises were carried out with the safety measures applied in cardiac rehabilitation: 3 sets of 10 rapid repetitions (with intensity at 75% of the 1 RM), rest time between the set of 1 or 2 min and avoiding the Valsalva maneuver. The progression of the weights and others extra exercises were adjusted by the physiotherapist during the sessions according to the patient [12].

The training program was progressively completed by treadmill and/or rowing machine (10 min at 90% of heart rate (HR) max).

Patients who had resting or exertional hypoxemia (generally oxygen saturation < 88%) were supplemented with O_2_.

The benefits obtained with the COVID-19 patients after rehabilitation were evaluated during the second CPET. The results were compared with the results of cardiac patients who had benefited from the same training program.

### 2.5. Statistical Analyses

Continuous variables were reported as mean and standard deviation (SD) after checking for normality of distribution by Fisher test.

The statistical analysis consisted of a repeated-measured analysis of variance (ANOVA), with modified Bonferroni post hoc tests when the F-ratio of the analysis of variance reached a *p* < 0.05 critical value.

Statistical analysis was performed using Statistica (Statistica version 10, StatSoft, Tulsa, OK, USA).

## 3. Results

The results of 25 patients who had suffered from moderate to severe forms of COVID-19 infection were analyzed and compared with the results of 25 cardiac patients who had completed a cardiopulmonary rehabilitation program in our center. In each group sixteen patients (64%) were male. In the COVID-19 patient group, the mean age was 61 years (±8) and the mean body mass index was 28.7 kg/m^2^ (±4.2) (Table 1).

Pulmonary function testing of these patients showed a mean forced expiratory volume in 1 s of 84% (±20) of predicted value with a lung vital capacity at 79% (±22) of predicted value. The diffusing capacity of the lung for carbon monoxide was reduced, at 52% (±11) of predicted value (Table 2).

At the beginning of their rehabilitation program COVID-19 patients presented, for a relatively high RER, a low physical capacity (16.8 ± 3.8 mL/kg/min, 71% of predicted value) with a reduced maximal workload (107 W ± 28, 70% of predicted value) (Table 3).

It was found that 19 patients among 25 had an altered exercise capacity, defined as a VO_2_ peak < 80% predicted (76%).

The ventilatory reserve at maximal exercise was still preserved for the majority of patients (37% on average for the VE/MVV). In the 19 patients with exercise limitation only 3 (15% of these patients) had consumed their ventilatory reserve (VE/MVV < 15%).

Mean SpO_2_ at rest was 98% (±2) and at maximal load was 95% (±4). Once more, in patients with exercise limitation, an oxygen desaturation during exercise (defined as a drop of initial saturation of 4% or more) was noted in four patients (21%). This desaturation was progressive and maximal at the end of the effort, as can be seen in patients with interstitial pneumopathy or with gas diffusion alteration.

Exercise hyperventilation measured by a higher VE/VCO_2_ slope was observed in some patients with a mean value at 37 (±6). Twelve patients with exercise limitation exhibited a VE/VCO_2_ slope above 35 (34%).

The mean HR was at 93/min (±15) at rest and 140/min (±7) at the end of the effort. During exercise no significative pathological response on ECG, in blood pressure nor in the oxygen pulse curve were observed.

The VT1 was measured at a mean value of 63% (±23) of VO_2_max predicted.

Data analysis identified 6 patients out of the 19 limited patients (31%) with a ventilatory limitation to exercise and 13 (69%) with a peripheral limitation.

On the one hand, after a mean of 23 (±5) sessions of rehabilitation, the maximal aerobic capacity (VO_2_max) and maximal workload of the COVID-19 patients increased by 18% and 26% to reach 84% and 88% of predicted values, respectively. Moreover, exercise hyperventilation decreased by 10%, as assessed by VE/VCO_2_ slope (Figure 1).

On the other hand, pulmonary intrinsic function improved following natural recovery. In fact, the vital capacity reached 91% of predicted values, the forced expiratory volume in 1 s was 94% but DLCO remained at only 63% (Table 2). However, even if the DLCO seems to remain impaired by the previous COVID-19 infection, patients were not limited by their lungs at maximal exercise.

Women and men were comparable in terms of initial VO_2_max and gain of VO_2_ after cardio-pulmonary rehabilitation (results no shown).

In the case of the cardiac patients, mean age was 62 years (±10) and the mean body mass index was 28.71 kg/m^2^ (Table 1).

The post-rehabilitation results of the cardiac patients were similar: maximal aerobic capacity (VO_2_max) and maximal workload increased by 22% and 25% to achieve 87% and 93% of predicted values, respectively (Figure 1). Exercise hyperventilation decreased by 10% (Table 3).

No incidents were reported during the CPET and the training sessions.

## 4. Discussion

CPET provides an objective assessment of exercise limitation and is included in the list of examinations of the European Respiratory Society (ERS)/American Thoracic Society (ATS) task force for the follow-up of COVID-19 patients [13].

The first CPET of this study confirmed exercise limitation after moderate to severe forms of COVID-19 infection. Belli et al. first described the reduced physical capacity of COVID-19 patients at the very beginning of the pandemic in 2020 [14].

It was also found that the exercise capacities of patients after COVID-19 infection were moderately impaired, as measured by VO_2_ max, which reached 71% of predicted values. As a reminder, the patients studied had suffered from moderate to severe forms of COVID-19 infection. In addition, our exercise evaluation was carried out relatively soon after discharge from hospital (4 weeks on average).

Motiejunaite et al. presented similar results concerning peak VO_2_ impairment 3 (±1) months after the initial diagnosis of COVID-19 infection [15]. As found in our study, effort limitation was predominant peripheral. Rinaldo et al. [16] and Raman et al. [17] also reported muscle deconditioning as the main mechanism of exercise intolerance 3 months after COVID-19 infection.

In their study, Medrinal et al. showed that the majority of COVID-19 ICU survivors had an acquired limb muscle weakness, and that the weakness was still present 4 weeks after discharge [18]. The evidence for COVID 19-induced muscle weakness seems clear, but the pathophysiological mechanisms involved are still under investigation. Histological sections of muscles post-COVID-19 present muscle fiber atrophy, metabolic alterations, and immune cell infiltration. The causes of muscle atrophy are multiple after COVID-19 infection: physical inactivity, systemic inflammation and cytokine storm, viral infiltration, malnutrition, hypoxemia and certain medications (e.g., dexamethasone). In addition, given the many neurological complications of COVID-19 infection, including peripheral nerve damage, it is likely that neuromuscular function may be impaired [19]. Future studies on this subject would be interesting. The role of resistance exercises, therefore, appears essential in the management of post-COVID-19 patients.

In this study, the proportion of patients with ventilatory limitation during exercise in the first CPET is higher than those found in the aforementioned studies. This can be explained by the fact that, in comparison, our evaluation was performed earlier following the acute infection, and it is known that pulmonary dysfunction tends to heal naturally over time [20].

Another common finding of all these studies is the elevated VE/VCO_2_ slope, suggesting a high incidence of inadequate exercise hyperventilation. Hyperventilation has been identified as a cause of exercise-induced dyspnea in PACS [21].

The recommendation of the European Respiratory Society- and American Thoracic Society is to propose a comprehensive rehabilitation program to COVID-19 survivors in need [8]. CPET is an essential tool in evaluating the need of rehabilitation and in establishing an efficient and safe program [22].

Soril et al. performed a rapid review of published literature over the rehabilitation proposed after COVID-19 infection. It revealed a wide variability in rehabilitation programs, but the study reported improvements in exercise capacity, pulmonary function, and/or quality of life for individuals who had been hospitalized due to an acute COVID-19 infection [23].

This study confirms that COVID-19 patients reap great benefits from cardio-pulmonary rehabilitation by measuring a significant improvement in their VO_2_ max. This VO_2_ gain obtained after rehabilitation is similar for COVID-19 survivors as for patients with ischemic cardiopathy.

The benefits of cardiac rehabilitation after a myocardial infarction and after coronary revascularization are clear [24]. In such circumstances cardiac rehabilitation decreases mortality, the risk of recurrence and improves quality of life [25,26].

As is the case for our cardiac patients, we offered High-Intensity Interval Training (HIIT) to the COVID-19 survivors, associated with resistance training, as combined training appears to be more efficient than either modality on its own [27].

HIIT is increasingly recommended in cardiac rehabilitation and has been endorsed by the American Heart Association Exercise since 2007 [28]. Guiraud et al. reviewed HIIT and cardiac rehabilitation literature and highlighted the ability of HIIT to improve peak oxygen consumption and achieve greater positive changes in cardiovascular risk factors, more successful than moderate continuous intensity training, without safety concerns and with better tolerance [29]. Moreover, in the randomized control trial of Wisløff et al., who studied the benefits of HIIT versus moderate continuous intensity training in post-infarction and heart failure patients, HIIT seemed to be more effective at improving peak oxygen consumption and was associated with lower levels of pro-brain natriuretic peptide. The HIIT group complained less during exercise and improved ejection fraction, stroke volume, and ventricular relaxation [30].

HIIT has been studied much less in pulmonary rehabilitation, when compared with cardiac rehabilitation. However, in Chronic Obstructive Pulmonary Disease (COPD) patients, HIIT does not appear to have inferior results to Moderate Continuous Training (MCT) for improving functional capacity [31]. Vogiatzis et al. evaluated skeletal muscle morphological and biochemical changes in patients with advanced COPD comparing HIIT versus MCT [32]. Skeletal muscle adaptations were clearly visible during both types of training, but HIIT was associated with a lower rate of negative symptoms during training sessions (dyspnea or leg pain, for example). These findings can lead us to understand why HIIT, in accordance with their CPET results, could be an effective pathway for the rehabilitation of post-COVID-19 patients.

## 5. Conclusions

Impairment of exercise capacity is common after COVID-19 infection and is often associated with persistent symptoms and altered quality of life. CPET is an essential tool in evaluating these patients, understanding their limitations during exercise and proposing adequate and safer cardio-pulmonary rehabilitation. A cardio-pulmonary rehabilitation program combining HIIT and resistance training significantly and similarly improves VO_2_ max in post-COVID-19 and cardiac patients (for whom the value of rehabilitation has been proven).

## Figures and Tables

**Figure 1 healthcare-10-02044-f001:**
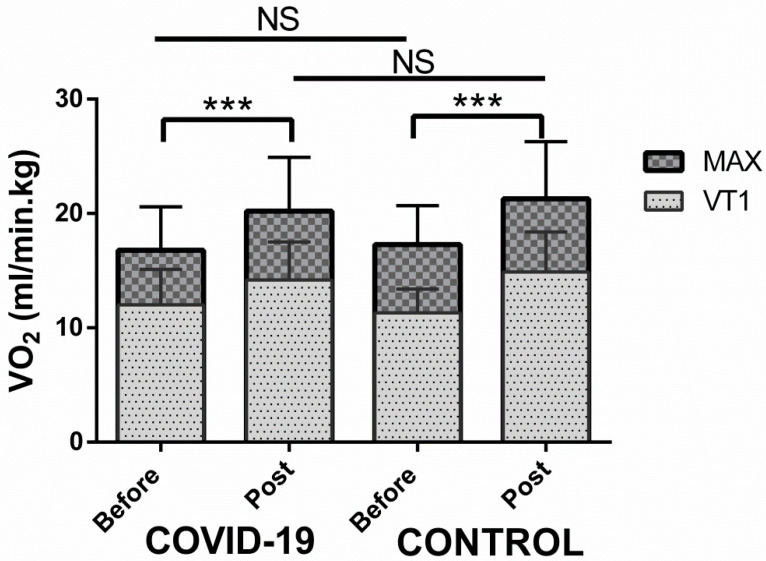
VO_2_ peak gain after cardiopulmonary rehabilitation in COVID-19 patients and in cardiac patients. The VO_2_ at ventilatory threshold 1 (VT1) and at the end of the exercise (MAX) were improved after 25 cardio-pulmonary rehabilitation sessions in COVID-19 patients and control group (cardiac patients). No statistical difference was found between these two groups. NS = non-significant; *** *p* < 0.001.

**Table 1 healthcare-10-02044-t001:** Study population and its characteristics.

	COVID-19 Patients		CONTROLS	
	Before	Post	Before	Post
Male/Female (*n* = 25)	16/9 (*n* = 25)		16/9 (*n* = 25)	
Age (years)	61 ± 8		62 ± 10	
Height (cm)	173 ± 11		170 ± 10	
Weight (kg)	86 ± 16	88 ± 15	81 ± 13	81 ± 13
BMI (kg/m^2^)	28.7 ± 4.2	29.2 ± 3.8	28.1 ± 3.8	28 ± 3.7
History of smoking	10		5	
Hypertension	12		18	
Diabetes	3		8	
Dyslipidemia	6		22 ^$$$^	
MEDICAL HISTORY				
Intensive Care Unit	13		Stenting	16
Mechanical ventilation	12		CABG	9
Noninvasive ventilation	16			
Oxygen supplementation	25			
Betablockers	5		20	

COVID-19 patients vs. control patients at the same stage of rehabilitation. ^$$$^
*p* < 0.001. If *p* value > 0.05: no symbol was added (no significant difference found).

**Table 2 healthcare-10-02044-t002:** Before rehabilitation and post-rehabilitation spirometry of COVID-19 patients.

	Before (*n* = 17)	After (*n* = 17)
LVC (L)	3.1 ± 0.7	3.5 ± 0.7 ***
LVC (% predicted value)	79 ± 22	91 ± 21 ***
FEV1 (L)	2.7 ± 0.8	2.9 ± 0.7
FEV1 (% predicted value)	84 ± 20	94 ± 19 *
DLCO (mL/min/mmHg)	14.2 ± 3.8	17.1 ± 4.4 ***
DLCO (% predicted value)	52 ± 11	63 ± 12 ***
KCO (mL/min/mmHg/L)	3.4 ± 0.8	3.6 ± 0.7
KCO (% predicted value)	82 ± 18	87 ± 18

LVC: lung vital capacity; FEV1: forced expiratory volume in 1 s; DLCO: diffusing capacity of the lung for carbon monoxide; KCO: rate of uptake of carbon monoxide from alveolar gas. * Before vs. post-rehabilitation. * *p* < 0.05; *** *p* < 0.001. If *p* value > 0.05: no symbol was added (no significant difference found).

**Table 3 healthcare-10-02044-t003:** Cardiopulmonary exercise testing before and after rehabilitation in 25 post-COVID-19 patients compared with 25 cardiac controls.

	COVID-19 Patients		CONTROLS	
	Before	Post	Before	Post
REST				
HR (bpm)	93 ± 15	89 ± 15	78 ± 13 ^$$^	75 ± 10 ^$$^
MBP (mmHg)	90 ± 10	93 ± 10	88 ± 12	89 ± 10
SpO_2_ (%)	98 ± 2	98 ± 1	98 ± 1	98 ± 1
VENTILATORY TRESHOLD 1				
VO_2_ (L/min)	1.0 ± 0.3	1.2 ± 0.6 ***	0.9 ± 0.2	1.2 ± 0.3 ***
VO_2_ (% predicted value)	51 ± 17	60 ± 19 ***	47 ± 12	61 ± 13 ***
Workload (W)	63 ± 23	86 ± 30 ***	65 ± 19	88 ± 30 ***
EqCO_2_	39 ± 6	36 ± 5 *	37 ± 4	35 ± 4 **
PetCO_2_ (mmHg)	36 ± 4	38 ± 4 *	37 ± 4	38 ± 4
HR (bpm)	117 ± 13	119 ± 13	98 ± 15 ^$$$^	103 ± 13 ^$$$^
MAXIMAL EFFORT				
Workload (W)	107 ± 28	137 ± 38 ***	111 ± 31	138 ± 45 ***
Workload (% predicted value)	70 ± 25	88 ± 27 ***	75 ± 20	93 ± 29 ***
VO_2_ (L/min)	1.4 ± 0.3	1.8 ± 0.5 ***	1.4 ± 0.3	1.7 ± 0.4 ***
VO_2_ (mL/min/kg)	16.8 ± 3.8	20.2 ± 4.7 ***	17.3 ± 3.4	21.3 ± 5 ***
VO_2_ (% predicted value)	71 ± 21	84 ± 22 ***	72 ± 15	87 ± 16 ***
RER	1.18 ± 0.10	1.16 ± 0.06	1.23 ± 0.10	1.20 ± 0.09
VE (L/min)	68 ± 14	78 ± 18 ***	71 ± 20	81 ± 24 ***
VE/MVV (%)	63 ± 13	83 ± 19 *	63 ± 9	76 ± 16 **
HR (bpm)	140 ± 17	144 ± 24	126 ± 20 ^$$^	134 ± 21 *
HR (% theoretical max HR)	88 ± 9	90 ± 13	80 ± 11 ^$$$^	85 ± 10 *^$^
MBP (mmHg)	115 ± 13	119 ± 13	117 ± 17	121 ± 19
SpO_2_ (%)	95 ± 4	95 ± 2	98 ± 1 ^$$^	98 ± 1 ^$$^
SLOPE				
VO_2_/W	9 ± 2	10 ± 1	9 ± 2	9 ± 2
FC/VO_2_	4.2 ± 1.4	3.8 ± 0.7	4.0 ± 1.3	3.7 ± 1.1
VE/VCO_2_	37 ± 6	34 ± 6	38 ± 7	35 ± 5

HR: heart rate; MBP: mean blood pressure; VO_2_: oxygen consumption; EqCO_2_: ventilatory equivalent in carbon dioxide; PetCO_2_: end tidal pressure in CO_2_; RER: respiratory exchange ratio; VE: ventilation; MVV: maximal voluntary ventilation; HRR: heart rate recovery. * Before vs. post-rehabilitation in the same group of patients * *p* < 0.05; ** *p* < 0.01; *** *p* < 0.001. ^$^ COVID-19 patients vs. control patients at the same stage of rehabilitation ^$^
*p* < 0.05; ^$$^
*p* < 0.01; ^$$$^
*p* < 0.001. If *p* value > 0.05: no symbol was added (no significant difference found).

## Data Availability

Not applicable.

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
