# Peer review of "Benefits of Cardio-Pulmonary Rehabilitation in Moderate to Severe Forms of COVID-19 Infection"

_healthcare, 2022, doi:10.3390/healthcare10102044_

Round 1
Reviewer 1 Report
Dear Editor and authors,
I would like to thank you for the opportunity to review this manuscript for your journal.
First of all, I would like to indicate that due to the subject matter, I consider it to be of great interest to the readers of the journal.
Next, I would like to make a series of questions for the authors to consider, since in my point of view they would be convenient to strengthen the presentation of this interesting research.
Firstly, I believe that since this is a pathology that is still being studied because the etiology (cytokine storm, hypercoagulability, pulmonary fibrosis, etc.) has not yet been established, this fact should be highlighted, since if it remains unknown it is difficult to establish a cause-effect relationship.
I believe it is necessary that at the end of the introduction the authors should explicitly state the general objective of the research.
The first time SpO2, HR, ECG, BP and the spirometer appear, although they are known, the device used (brand, model, city, country) should appear.
Line 112. "w "arm should be in capital letters.
Describe if the HIIT was performed with or without music (because of the effect that this can have on the physiological level).
The spirometry process should be described (I understand that it will be the standard, but it is convenient to describe it).
In table 1 change Before by Pre.
I do not know if there is an error in the wording, but I understand that there are 25 COVID subjects and the table shows 17 --> why this difference? experimental death? error?
From this consideration, it would be convenient to design a flow chart.
Neuromuscular function has not been measured, what role do the authors consider it may play in the results obtained? As they describe, due to bed rest, pathology and pharmacological treatment, sarcopenia may induce a sarcopenia that would influence the results.
In figure 1 a legend should be added to know the meaning of the abbreviations
It should be described whether a familiarization period or intensity progression was performed prior to the intervention.
Because of the effects that COVID-19 has on the respiratory system it would be worthwhile to detail whether the subjects had received respiratory physiotherapy.
Did they measure lactate during CPET? would be a very interesting parameter.
Due to the muscular effect, it would be worth indicating that it would be interesting in future lines of research to add the analysis of the neuromuscular performance and the effects on this type, for example markers of muscular damage, stiffness, etc.
Best of luck
Author Response
Dear Editor and authors,
I would like to thank you for the opportunity to review this manuscript for your journal.
First of all, I would like to indicate that due to the subject matter, I consider it to be of great interest to the readers of the journal.
Next, I would like to make a series of questions for the authors to consider, since in my point of view they would be convenient to strengthen the presentation of this interesting research.
We sincerely thank the reviewer for the critical review and we hope to answer appropriately.
Firstly, I believe that since this is a pathology that is still being studied because the etiology (cytokine storm, hypercoagulability, pulmonary fibrosis, etc.) has not yet been established, this fact should be highlighted, since if it remains unknown it is difficult to establish a cause-effect relationship.
We added it in the introduction and in the discussion:
- “The pathophysiological mechanisms are still under investigation and the causes of a negative progression to ARDS are not yet fully established [3].” Line 37
- “The evidence for covid 19-induced muscle weakness seems clear but the pathophysiological mechanisms involved are still under investigation.” Line 290
I believe it is necessary that at the end of the introduction the authors should explicitly state the general objective of the research.
Thanks for this comment. We added the aim at the end of the introduction to clarify our subject of study (page 2, line 62): « The aim of this study was to evaluate the benefits of cardio-pulmonary rehabilitation with high intensity interval training, on severe to moderate COVID-19 patients. These benefits were then compared to those obtained with the patients suffered from ischemic cardiopathy, for whom the value of cardiopulmonary rehabilitation is well established ».
The first time SpO2, HR, ECG, BP and the spirometer appear, although they are known, the device used (brand, model, city, country) should appear.
Thank you for this observation. We added all these precisions in the Methodology part of the article (page 2 and 3).
Line 112. "w "arm should be in capital letters.
We removed the point to correct this mistake.
Describe if the HIIT was performed with or without music (because of the effect that this can have on the physiological level).
The training session were done without music, we added this precision on the line 153.
The spirometry process should be described (I understand that it will be the standard, but it is convenient to describe it).
We added one paragraph in the Methology and specified the guidelines followed to make the measurements.
In table 1 change Before by Pre.
Done.
I do not know if there is an error in the wording, but I understand that there are 25 COVID subjects and the table shows 17 --> why this difference? experimental death? error?
From this consideration, it would be convenient to design a flow chart.
Thank you for this comment. It was indeed not clearly explained. We specified the explanation page 2 in the paragraph Study design. Actually, it is a retrospective study and unfortunately only 17 patients on de 25 COVID-19 patients had a pulmonary function testing before and after rehabilitation.
Neuromuscular function has not been measured, what role do the authors consider it may play in the results obtained? As they describe, due to bed rest, pathology and pharmacological treatment, sarcopenia may induce a sarcopenia that would influence the results.
Our aim was to evaluate the benefits of the cardio-pulmonary rehabilitation in term of VO2 max gain since it is correlate with the quality of life and the mortality. Furthermore, all our patients were autonomous when walking. Specific testing of neuromuscular function would therefore not have been relevant in the management of the patients.
In figure 1 a legend should be added to know the meaning of the abbreviations
Done.
It should be described whether a familiarization period or intensity progression was performed prior to the intervention.
Done on the paragraph Study design.
Because of the effects that COVID-19 has on the respiratory system it would be worthwhile to detail whether the subjects had received respiratory physiotherapy.
COVID-19 patients were addressed one month after discharge. Rehabilitation must be patient-centred and tailored to individual patient needs. (“The Stanford Hall consensus statement for post-COVID-19 rehabilitation” in Br J Sports Medicine 2020). No patient was referred or needed respiratory physiotherapy. In respiratory physiotherapy we mean alveolar recruitment and exercise at high volume and low output.
Did they measure lactate during CPET? would be a very interesting parameter.
Unfortunately we did not perform a blood sample lactate assay (invasive measurement) but it would have been very interesting. We did have an indirect measurement of aerobic and anaerobic energy metabolism used (and therefore of lactic acid production) via the SV1 and the RER.
Due to the muscular effect, it would be worth indicating that it would be interesting in future lines of research to add the analysis of the neuromuscular performance and the effects on this type, for example markers of muscular damage, stiffness, etc.
This topic is indeed very interesting and should be the subject of future studies, I mentioned it in the discussion line 300-305.
Reviewer 2 Report
1. The title should not contain abbreviations not commonly known (HILT). It is not known why it is annotated that it is a retrospective study.
2. Objective does not contain a scientific dilemma to be solved. It is only about the benefits after rehabilitation, which are usually obvious.
3. There is no separation of groups according to gender, which must interfere with the results, for example in terms of the effects obtained from resistance exercises.
4. Lack of description of resistance exercises and their dosing, as well as interval exercises.
5. There are no references to the category of statistical significance in the descriptions of the results. What does it mean that the results were similar?
6. The conclusions are trivial and not very revealing that rehabilitation improves the physical efficiency of the respondents and is effective and safe.
Author Response
Dear reviewer,
First of all, I would like to thank you for the attention you have paid to this article. We have done our best to respond to each of your comments correctly.
- The title should not contain abbreviations not commonly known (HILT). It is not known why it is annotated that it is a retrospective study.
Thanks for this comment: we have indeed simplified the title of the article.
- Objective does not contain a scientific dilemma to be solved. It is only about the benefits after rehabilitation, which are usually obvious.
Indeed, experts recommend rehabilitation after Covid-19 infection. However, there is a lack of data in the literature on the timing of this rehabilitation, on the training modalities, on the expected benefits (quantitative measurement) and on the risks of possible incidents. A combined rehabilitation program (aerobic and resistance exercise) which is clearly recommended for cardiac patients, could be a good training model in view of the COVID-19 patients' symptoms, CPET results and muscle exercise limitation. The aim was to measure the potential benefits of cardiac rehabilitation training modalities transposed to a COVID-19 population which is currently understudied. We have chosen to use the CPET to objectively evaluate the benefits obtained. We propose here a rehabilitation program based on an initial assessment and follow-up with CPET as recommended for cardiac patients. CPET allows us to establish a clear and precise training program while reinforcing the safety during the training sessions.
- There is no separation of groups according to gender, which must interfere with the results, for example in terms of the effects obtained from resistance exercises.
Effectively, sex differences in adaptations to resistance training are apparent in older adults (> 50 years-old). However, the interpretation of sex-dependant adaptations to resistance training is heavily influenced by the presentation of the results in either an absolute or relative context. (Jones, M. D., Wewege, M. A., Hackett, D. A., Keogh, J., & Hagstrom, A. D. (2021). Sex Differences in Adaptations in Muscle Strength and Size Following Resistance Training in Older Adults: A Systematic Review and Meta-analysis. Sports medicine (Auckland, N.Z.), 51(3), 503–517. https://doi.org/10.1007/s40279-020-01388-4).
Here, the subjects were matched 1 by 1 according to sex, age, weight, height and initial exercise capacity. The potentially increased gain of males compared to females seems to be ruled out by our methodology.
- Lack of description of resistance exercises and their dosing, as well as interval exercises.
Thank you for this comment. We have strengthened the description of the patient training (especially for the resistance exercises) in paragraph 2.2 of the methodology, in the cardiopulmonary rehabilitation section.
“For the resistance exercises, the one repetition maximal (1 RM) was determined after the familiarization session on the vertical traction and triceps machine for upper limb and the leg press, the leg extension, the leg curl for lower limb were the main machines used. The resistance exercises were carried out with the safety measures applied in cardiac rehabilitation: 3 sets of 10 rapid repetitions (with intensity at 75% of the 1 RM), rest time between the set of 1 or 2 minutes and avoiding the Valsalva maneuver. The progression of the weights and others extra exercises were adjusted by the physiotherapist during the sessions according to the patient.”
Reference: Bjarnason-Wehrens, B., Schwaab, B. Resistance Training in Patients With Coronary Artery Disease, Heart Failure, and Valvular Heart Disease: A REVIEW WITH SPECIALEMPHASIS ON OLD AGE, FRAILTY, AND PHYSICAL LIMITATIONS. Journal of cardiopulmonary rehabilitation and prevention, 2022, 42(5), 304–315. https://doi.org/10.1097/HCR.0000000000000730.
- There are no references to the category of statistical significance in the descriptions of the results. What does it mean that the results were similar?
We used the classical method to the category of statistical significance (p<0.05; p<0.01; p<0.001). The sign "*" was used for the comparison before versus post rehabilitation and the sign "$" was used for the comparison COVID-19 patients versus control/cardiac patients. We have not put anything next to the values if there was no significant difference so as not to make the tables more cumbersome. Thanks to your remark, we have corrected the legend of table 1.
- The conclusions are trivial and not very revealing that rehabilitation improves the physical efficiency of the respondents and is effective and safe.
We have taken your opinion into account and have tried to work the conclusion to make our results clearer (line 363-368).

Round 2
Reviewer 2 Report
The explanation for point 3 of the previous review is unconvincing.
To point 4, please add a description of the dosing of loads and the method of achieving the record (1RM).
Please add an explanation to point 5 - whether the changes achieved in the groups and between the groups were statistically significant.
Author Response
Once again, we would like to thank you for your time and feedback.
- The explanation for point 3 of the previous review is unconvincing.
The aim of our study was not to compare the effects of cardiopulmonary rehabilitation after Covid 19 infection between women and men.
However, if we look at the data a little more closely:
During the first CPET, the VO2 max values were higher in men in absolute terms: 1.29 L for women and 1.5 L for men, with a p-value of 0.02. However, the VO2 max values are comparable when indexed to weight: 16.6 ml/min/kg in women and 17.5 in men (p-value = 0.40 and therefore no significant difference).
Men ventilated more (76 L/min for men and 57 L/min for women) but ventilatory reserves were identical (72% in women and 69% in men with a p-value of 0.48).
The secondary data were not statistically different.
Thus, the physical capacities of the two groups before rehabilitation were comparable.
For the physical capacity gains assessed in the second CPET, women gained an average of 0.250 L/min of absolute VO2 max and men 0.383 L/min. These gains were therefore not significantly different either with a p value of 0.054.
By analysing these values, we can state that men and women improved in the same way.
We added one sentence in the results paragraph:
“Women and men were comparable in term of initial VO2max and gain of VO2 after cardio-pulmonary rehabilitation (results no shown).” Line 264.
- To point 4, please add a description of the dosing of loads and the method of achieving the record (1RM).
“To determine the one repetition maximum, we asked the patient to do the maximum number of repetitions with a weight that we gradually increased by 2 kilos for the upper limbs and 5 kilos for the lower limbs. The one repetition maximal was the load with which the patient was able to do only one repetition.” Line 158.
- Please add an explanation to point 5 - whether the changes achieved in the groups and between the groups were statistically significant.
In the results paragraph, we completed the explanation of our statistical results after each table.
“*Before vs. post-rehabilitation in the same group of patients *p<0.05; ** p<0.01; *** p<0.001.
$COVID-19 patients vs. control patients at the same stage of rehabilitation $p<0.05; $$ p<0.01; $$$ p<0.001.
If P value > 0.05: no symbol was added (no significant difference found).” Line 232.
